# Association between Fat-Free Mass and Brain Size in Extremely Preterm Infants

**DOI:** 10.3390/nu13124205

**Published:** 2021-11-24

**Authors:** Christoph Binder, Julia Buchmayer, Alexandra Thajer, Vito Giordano, Victor Schmidbauer, Karin Harreiter, Katrin Klebermass-Schrehof, Angelika Berger, Katharina Goeral

**Affiliations:** 1Comprehensive Center for Pediatrics, Department of Pediatrics and Adolescent Medicine, Division of Neonatology, Intensive Care and Neuropediatrics, Medical University of Vienna, 1090 Vienna, Austria; julia.buchmayer@muv.ac.at (J.B.); alexandra.thajer@muv.ac.at (A.T.); vito.giordano@muv.ac.at (V.G.); karin.harreiter@muv.ac.at (K.H.); katrin.klebermass-schrehof@muv.ac.at (K.K.-S.); angelika.berger@muv.ac.at (A.B.); katharina.goeral@muv.ac.at (K.G.); 2Department of Radiology, Division of Neuroradiology and Musculoskeletal Radiology, Medical University of Vienna, 1090 Vienna, Austria; victor.schmidbauer@muv.ac.at

**Keywords:** air displacement plethysmography, body composition, brain, cerebral, magnetic resonance imaging, neonate, preterm

## Abstract

Postnatal growth restriction and deficits in fat-free mass are associated with impaired neurodevelopment. The optimal body composition to support normal brain growth and development remains unclear. This study investigated the association between body composition and brain size in preterm infants. We included 118 infants born <28 weeks of gestation between 2017–2021, who underwent body composition (fat-free mass (FFM) and fat mass (FM)) and cerebral magnetic resonance imaging to quantify brain size (cerebral biparietal diameter (cBPD), bone biparietal diameter (bBPD), interhemispheric distance (IHD), transverse cerebellar diameter (tCD)) at term-equivalent age. FFM Z-Score significantly correlated with higher cBPD Z-Score (rs = 0.69; *p* < 0.001), bBPD Z-Score (rs = 0.48; *p* < 0.001) and tCD Z-Score (rs = 0.30; *p* = 0.002); FM Z-Score significantly correlated with lower brain size (cBPD Z-Score (rs = −0.32; *p* < 0.001) and bBPD Z-Score (rs = −0.42; *p* < 0.001). In contrast weight (rs = 0.08), length (rs = −0.01) and head circumference Z-Score (rs = 0.14) did not. Linear regression model adjusted for important neonatal variables revealed that FFM Z-Score was independently and significantly associated with higher cBPD Z-Score (median 0.50, 95% CI: 0.59, 0.43; *p* < 0.001) and bBPD Z-Score (median 0.31, 95% CI: 0.42, 0.19; *p* < 0.001); FM Z-Score was independently and significantly associated with lower cBPD Z-Score (median −0.27, 95% CI: −0.42, −0.11; *p* < 0.001) and bBPD Z-Score (median −0.32, 95% CI: −0.45, −0.18; *p* < 0.001). Higher FFM Z-Score and lower FM Z-scores were significantly associated with larger brain size at term-equivalent age. These results indicate that early body composition might be a useful tool to evaluate and eventually optimize brain growth and neurodevelopment.

## 1. Introduction

Premature infants are at increased risk for adverse neurological outcomes [1]. Appropriate nutritional management is important to support normal growth and brain development [2,3]. Several studies demonstrated that an inadequate nutritional intake is associated with growth failure and smaller brain size [4,5]. It is well accepted that brain size is an important marker for neurodevelopmental impairment in premature infants [6,7]. The major nutritional goal is to achieve an optimum nutritional intake to avoid growth restriction and neurodevelopmental delay [8]. In addition to nutritional management, multiple factors influence growth and body composition, including gestational age at birth, birth weight, brain injury, inflammation and infection, and other neonatal morbidities [9]. Several studies demonstrated that undernutrition and protein deficiency are associated with growth failure [4,8]. However, the appropriate nutritional management in extremely preterm infants has not been fully investigated thus far. Growth failure is associated with an impaired neurodevelopment, and growth and is usually monitored by anthropometric parameters including weight, length and head circumference and provides information about the quantity, but not about the quality of growth, especially fat-free mass (FFM), or lean mass, and fat mass (FM) [10]. Body composition is a useful and valuable tool to assess the nutritional status and the nutritional management of extremely preterm infants [10,11]. However, the optimal body composition to support normal brain growth and development in this high-risk patient collective still remains unclear. Studies provide evidence that FFM is positively associated with neurodevelopmental outcome [12]. The aim of this study was to evaluate the association between body composition (FFM and FM), anthropometric parameters (weight, length and head circumference) and brain size at term-equivalent age.

## 2. Materials and Methods

This retrospective study was conducted at a level IV neonatal intensive care unit at the Medical University of Vienna. Preterm infants born <28 weeks of gestation between September 2017 and March 2021 were eligible. Only infants who underwent both air displacement plethysmography (Pea Pod) to determine body composition, and cerebral magnetic resonance imaging (cMRI) to quantify brain size, at term-equivalent age were included in the analyses. In our hospital, both body composition and cMRI are performed at term-equivalent age as part of our standard clinical practice in all preterm infants born <28 weeks of gestation. Exclusion criteria were major congenital anomalies, brain malformations, metabolic disorders and chromosomal abnormalities. In addition, patients with body composition or cMRI ≥ 43 weeks were excluded, as normative data are not available thereafter.

### 2.1. Body Composition

Preterm infants underwent non-invasive air displacement plethysmography (Pea Pod, COSMED, CA, USA) to determine body composition at term-equivalent age. The device is based on a two-compartment model of body composition including FFM and FM and uses the inverse relation between pressure and volume to derive body volume for a subject [13]. The accuracy and reliability of the device has been reported previously [14]. FFM and FM Z-Scores were calculated according to the recently published sex and gestational-age specific reference charts for premature infants up to 6 months of age [10]. FFM and FM percentage as well as kilograms are reported.

### 2.2. Neuroimaging

Routine cMRI was performed using a 1.5 Tesla scanner without sedation, adapted to the protocol by Woodward et al. [7]. The local neonatal standard protocol includes conventional T1- and T2-weighted images in three orthogonal planes, diffusion-weighted imaging (b-values of 0 and 700 s/mm^2^) and susceptibility-weighted imaging. Brain measurements were performed by two independent investigators (KG, JB), who were blinded to the clinical history. 

One-dimensional measurements were evaluated by obtaining cerebral biparietal diameter (cBPD), bone biparietal diameter (bBPD), interhemispheric distance (IHD), and transverse cerebellar diameter (tCD) in coronar view (Figure 1) [15,16]. BPD and IHD measurements were taken at the level of the temporal horns of the lateral ventricles. Arteria basilaris and bilateral cochlea were used as landmarks. cBPD corresponded to the greatest transverse brain width at the mentioned level, while bBPD was defined as the maximum diameter between the internal margin of the skull. IHD was measured as the horizontal distance between the internal edges of the superior frontal gyri, directly above the cingular sulcus, at an equal distance from corpus callosum and vertex [16]. tCD was obtained as the maximum diameter of the vermis on a posterior coronal slice at the level of the ventricular atria. For each of the mentioned measurements Z-Scores were calculated by comparing the obtained measurement to normative data of healthy age-matched controls published by either Garel (fetal cMRI data; comparison of all cMRI performed <38 weeks of gestation) [15] or Nguyen Te Tich et al. (postnatal cMRI; comparison of all cMRI performed at term-equivalent age (38–42 weeks of gestation) [16].

The presence of impaired brain growth, corresponding to the small cBPD or increased IHD brain pattern, was evaluated. Definitions were adopted according to Kidokoro et al. [17]. Each of the mentioned patterns was considered to be present if the obtained measurement deviated >2 standard deviations from normative values [17].

### 2.3. Statistics

Demographic data and descriptive statistics were expressed as median and interquartile range (IQR) and frequency distribution. Weight, length, head circumference, FFM and FM, and were transformed into Z-Scores using the LMS method, based on sex and gestational age specific growth charts [10,15,16]. Brain measurement (cBPD, bBPD, IHD and tCD) Z-Scores were created as described above. Pearson correlation test was used to assess the correlation between body composition (FFM and FM Z-Scores) and brain metric measurements (cBPD, bBPD, IHD and tCD Z-Scores). Lineal regression model was applied to examine the association between the outcome parameters of body composition with adjustment for the covariates, sex [18], gestational age at birth [19], birth weight Z-Score [20], length of parenteral nutrition [21], the illness severity score (SNAPPE-II) [22], and brain metric measurement (cBPD, bBPD, IHD, and tCD) Z-Scores. Estimated mean and adjusted mean differences with 95% confidence intervals (CI) were reported. Intra- as well as interrater agreement were assessed in 10% of patients. Intraclass correlation coefficients were calculated using a 2-way random model for absolute agreement and interpreted according to the strength of agreement scale of Brennan and Silman. Data were analyzed using SPSS, version 27 for Mac (IBM Corp, Armonk, New York, NY, USA). A *p*-value of <0.05 was considered statistically significant.

## 3. Results

A total of 139 extremely preterm infants underwent body composition and cMRI. After exclusion of 21 patients with body composition or cMRI ≥ 43 weeks’, a total of 118 extremely preterm infants were available for final analysis. Baseline characteristics, nutritional data and short-term outcome parameters are shown in Table 1. Median gestational age at birth was 26.1 weeks (IQR: 24.5, 27.0) and median birth weight 770 g (IQR: 645, 923). Median weight Z-Score decreased from 0.1 (IQR: −0.6, 0.5) to −1.0 (IQR: −1.6, −0.5) from birth to term-equivalent age. Median length Z-Score declined from 0.1 (IQR: −0.5, 0.7) to −1.2 (IQR: −2.1, −0.4) and head circumference Z-Score from 0.0 (IQR: −0.5, 0.7) to −0.7 (IQR: −1.4, −0.2) until term-equivalent age.

Anthropometric and body composition measurements (median age at measurement: 41.4 weeks, IQR: 40.0, 44.5) and brain size data (median age at cMRI: 37.4 weeks, IQR: 36.4, 38.6) are shown in Table 2. Median FFM percentage was 79.6% (IQR: 75.1, 83.1), and median FM percentage was 20.4% (IQR: 16.9, 24.9). Median FFM Z-Score was −1.8 (IQR: −2.7, −0.7) and median FM Z-Score 1.1 (IQR: 0.4, 1.8). Median length of parenteral nutrition was 26 days (IQR: 17, 36) in the study cohort, and around 66% of infants were fed exclusively mother’s milk at discharge. Median cBPD Z-Score and median bBPD Z-Score were below two standard deviations (cBPD: −2.3 (IQR: −2.8, −1.8) and bBPD: −2.2 (IQR: −2.9, −1.7). Median IHD Z-Score was 0.3 (IQR: −0.3, 0.8), median tCD Z-Score −0.7 (IQR: −2.1, 0.0), both in normative ranges. Impaired brain growth was present in 73% (86/118) of all patients: 68% (80/118) showed a small cBPD pattern, 9% (10/118) an increased IHD pattern and 3% (4/118) impaired brain growth according to both patterns. Intra- and interrater agreement was classified as very good (≥0.81) according to Brennan and Silman.

FFM Z-Score significantly correlated with higher cBPD Z-Score (r_s_ = 0.69; *p* < 0.001), bBPD Z-Score (r_s_ = 0.48; *p* < 0.001) and tCD Z-Score (r_s_ = 0.30; *p* = 0.002) (Figure 2). FFM Z-Score demonstrated no correlation with IHD Z-Score (r_s_ = 0.1; *p* = 0.19). FM Z-Score significantly correlated with lower cBPD Z-Score (r_s_ = −0.32; *p* < 0.001) and bBPD Z-Score (r_s_ = −0.42; *p* < 0.001). FM Z-Score demonstrated no correlation with IHD Z-Score (r_s_ = 0.14; *p* = 0.12) and tCD Z-Score (r_s_ = 0.12; *p* = 0.16). Weight (r_s_ = 0.08; *p* = 0.45), length (r_s_ = −0.01; *p* = 0.94) and head circumference (r_s_ = 0.14; *p* = 0.17) Z-Scores at term-equivalent age did not correlate with brain size parameters.

Linear regression model, adjusted for gestational age at birth, birth weight Z-Score, sex, length of parenteral nutrition, and SNAPPE-II Score revealed that FFM Z-Score was independently and significantly associated with higher cBPD Z-Score (median 0.50, 95% CI: 0.59, 0.43; *p* < 0.001) and bBPD Z-Score (median 0.31, 95% CI: 0.42, 0.19; *p* < 0.001). FM Z-Score was independently and significantly associated with lower cBPD Z-Score (median −0.27, 95% CI: −0.42, −0.11; *p* < 0.001) and bBPD Z-Score (median −0.32, 95% CI: −0.45, −0.18; *p* < 0.001). FFM and FM Z-Scores had no association with IHD Z-Score (*p* = 0.06, *p* = 0.07) and tCD Z-Score (*p* = 0.21, *p* = 0.25), respectively.

Length of parenteral nutrition had a significant negative association with FFM (median −0.06 Z-Score, 95% CI: −0.11, 0.03; *p* < 0.001) and cBPD (median −0.05 Z-Score, 95% CI: −0.10, −0.03; *p* = 0.037). Gestational age at birth (*p* = 0.52), birth weight Z-Score (*p* = 0.33), sex (*p* = 0.06) and SNAPPE-II Score (*p* = 0.07) had no significant association with brain size parameters. By term-equivalent age, weight as well as length and head circumference Z-Scores did not correlate with brain size Z-Scores.

## 4. Discussion

This study showed a significant positive association between FFM Z-Score, FM Z-Score and brain size Z-Scores in extremely preterm infants at term-equivalent age. In contrast, weight, length and head circumference Z-Scores did not significantly correlate with brain size. This study confirms the hypothesis that the FFM Z-Score is rather a benchmark for optimal brain growth than anthropometric parameters.

It is well accepted that an adequate nutritional management and status are essential to support normal preterm brain growth and neurodevelopment [2,3]. Several studies demonstrated that postnatal growth restriction and impaired brain growth are associated with long-term neurodevelopmental impairment [4,5]. Growth is usually monitored by anthropometric parameters, but body composition provides additional information on the nutritional status of an infant, including FFM and FM [14]. However, the optimal body composition to support normal brain growth and development in premature infants still remains unclear. Studies provide evidence that deficits in FFM (lean mass) during growth are associated with neurodevelopmental impairment [12,23]. Previous studies concluded that adiposity and higher FM might be linked to smaller brain size [4,24]. In contrast, the study by Paviotti et al. [25] showed an association between higher FM and larger cerebellar volumes.

In our cohort, anthropometric parameters (weight, length, head circumference Z-Scores) were not significantly associated with brain size Z-Scores, while FFM Z-Score was significantly association with larger brain size. The association between body composition and brain size was evaluated with conflicting results only by a few studies thus far, all with considerably smaller sample sizes [24,25,26]. Vasu et al., found a significant negative association of body fat (deep subcutaneous abdominal adipose tissue volume) measured by whole-body MRI with brain size [24]. The sample size was relatively small (*n* = 22) in comparison to our study (*n* = 118), but results were consistent [24]. Paviotti et al. showed that FFM and FM are associated with higher cerebellar volume calculated by 3D reconstruction at term-equivalent age [25]. In that study, brain size was evaluated by cerebellar volume and no additional brain regions were analyzed. We did not measure cerebellar volume but tCD in our study; but both studies underline the hypothesis that FFM is a potential marker to support normal brain size [25]. Contrary to the study by Paviotti et al., FM Z-Score was negatively associated with brain size in our study [25]. Patients in Paviotti’s study were substantially older at birth (median gestational age: 26.1 versus 29.4 weeks’) and rarely showed growth restriction. We hypothesize that infants in our study were more immature and vulnerable for malnutrition, leading to FM loss and impaired brain size. In the study by Bell et al., FFM was independently and significantly association with larger brain size at term-equivalent age, consistent with our results [26]. In comparison to the study by Bell et al., infants in our study were substantially younger (median gestational age: 26 + 1 versus 29 + 1 weeks) and smaller at birth (median weight: 770 versus 1088 g) [26]. In addition, weight Z-Score and FFM percentage were lower in our study compared to the study by Bell et al. (median weight Z-Score: −1.0 versus −0.6; median FFM percentage: 79.6 versus 82.3%).

Several factors influence body composition including gestational age at birth, birth weight, brain injury, inflammation and infection, other neonatal morbidities and especially the early nutritional management [8,9]. Studies showed that an increased energy and protein intake is associated with an improved head growth and larger FFM [27]. FFM may be an indicator of protein accretion and brain growth. We hypothesize that infants with higher FFM in our study cohort received an adequate nutritional management with an optimum protein and energy supply. Further interventional studies are necessary to examine the impact of early nutrition on body composition, brain size and most importantly neurodevelopment. Previous studies investigating the association between FFM gain and neurodevelopment showed conflicting results [12,28]. Body composition and especially faster FFM gain at term-equivalent age were not linked to better outcome, while FFM at term-equivalent age has been shown to be the best marker for neurodevelopment [8,28]. These data indicate that the window for an optimal nutritional management is small and early adequate nutrition is essential to support normal neurodevelopment.

Additionally, we found that length of parenteral nutrition is significantly associated with FFM and brain size. Several studies demonstrated that premature infants are at high risk for malnutrition due to long-term parenteral nutrition [8,21,29]. Prolonged parenteral nutrition may lead to energy and protein deficits and several studies demonstrated that adequate protein supply is important for linear growth and FFM gain [21,30]. The major nutritional goal in preterm infants is to achieve adequate nutritional intake to avoid growth restriction, which is associated with impaired neurodevelopment [31]. In our study, infants received parenteral nutrition for a relatively long time compared to other reports in the literature (median 26 versus 17 days) [26], where the nutritional management and especially the time on parenteral nutrition had no significant impact on body composition or brain size. We hypothesize that the longer time on parenteral nutrition in our study cohort, can be explained by the high-risk study cohort of only extremely preterm infants (median gestational age: 26 + 1 weeks, median birth weight: 770 g). This fact may have led to an energy and protein deficiency resulting in lower FFM.

One limitation of the present study is that data on long-term neurodevelopment is not yet available. Follow-up data might be helpful and will be evaluated in this cohort in the future. A strength of our study is the relatively large sample size and the use of Z-Scores for anthropometric parameters, body composition as well as cMRI measurements. To the best of our knowledge, this is the largest cohort of extremely preterm infants who underwent body composition and cMRI at term-equivalent age.

## 5. Conclusions

This study found an association between higher FFM Z-Score and larger cMRI measured brain size at term-equivalent age. In contrast, a higher FM Z-Score was associated with smaller brain size. Body composition measurements might be a useful tool to evaluate and eventually optimize brain growth and further neurodevelopment.

## Figures and Tables

**Figure 1 nutrients-13-04205-f001:**
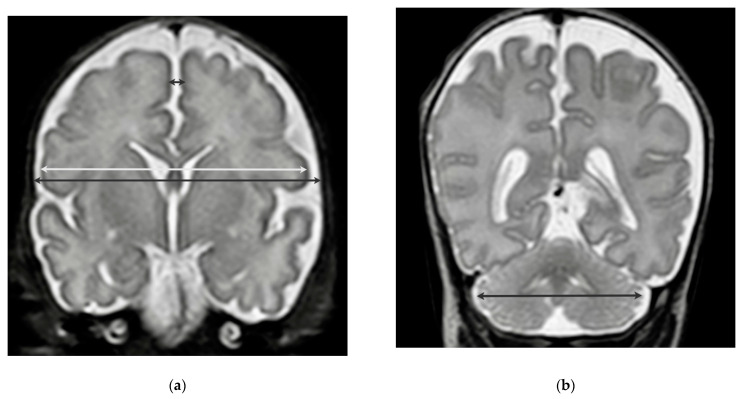
Measurements of (**a**) cerebral biparietal diameter (cBPD, light grey), bone biparietal diameter (bBPD, dark grey), interhemispheric distance (IHD, black) and (**b**) transverse cerebellar diameter (tCD, black).

**Figure 2 nutrients-13-04205-f002:**
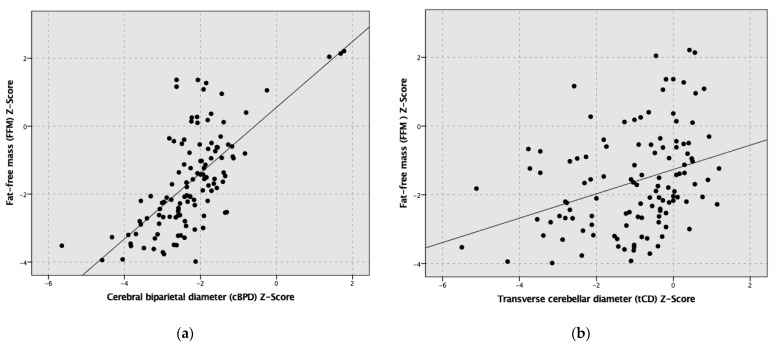
Scatterplots of fat-free mass Z-Score and (**a**) cerebral biparietal diameter Z-Score; (**b**) transverse cerebellar diameter Z-Score at term-equivalent age (*n* = 118).

**Table 1 nutrients-13-04205-t001:** Baseline characteristics.

**Patient group (*n* = 118)**	
Male, % (*n*)	61.0 (72/118)
Gestational age, weeks	26.1 (24.5, 27.0)
**Anthropometry at birth**	
Weight, gram	770 (645, 923)
Length, cm	33.0 (31.0, 35.0)
Head circumference, cm	23.5 (22.0, 25.0)
Weight, Z-Score	0.1 (−0.6, 0.5)
Length, Z-Score	0.1 (−0.5, 0.7)
Head circumference, Z-Score	0.0 (−0.5, 0.7)
Small for gestational age (<10th percentile), % (*n*)	10.2 (12/118)
Cesarean delivery, % (*n*)	84.7 (100/118)
APGAR Score, 5 min	9 (8, 9)
APGAR Score, 10 min	9 (9, 9)
Umbilical artery, pH	7.32 (7.28, 7.36)
SNAPP-II Score	9.0 (0, 9)
Necrotizing enterocolitis (stage ≥ 2), % (*n*)	5.1 (6/118)
Intraventricular hemorrhage (stage > 2), % (*n*)	10.2 (12/118)
Treatment for posthemorrhagic hydrocephalus, % (*n*)	3.4 (4/118)
Culture proven septicemia, % (*n*)	17.8 (21/118)
Parenteral nutrition, days	26 (17, 36)
Exclusively mother’s milk at discharge, % (*n*)	66.1 (78/118)

Data are % (*n*) or median (IQR—interquartile range) as appropriate.

**Table 2 nutrients-13-04205-t002:** Body composition and brain size parameters at term equivalent age.

Patient Group (*n* = 118)	
**Body composition parameters**	
Postmenstrual age, weeks	41.4 (40.0, 44.5)
FFM, percentage	79.6 (75.1, 83.1)
FM, percentage	20.5 (16.9, 24.9)
FFM, gram	2716 (2300, 3141)
FM, gram	702 (515, 947)
FFM, Z-Score	−1.8 (−2.7, −0.7)
FM, Z-Score	1.1 (0.4, 1.8)
Weight, gram	3473 (2920, 4023)
Weight, Z-Score	−1.0 (−1.6, −0.5)
Length, cm	50.0 (48.4, 53.6)
Length, Z-Score	−1.2 (−2.1, −0.4)
Head circumference, cm	35 (33, 36)
Head circumference, Z-Score	−0.7 (−1.4, −0.2)
**Neuroimaging parameters**	
Postmenstrual age, weeks	37.4 (36.4, 38.6)
cBPD, mm	71.1 (68.9, 74.1)
bBPD, mm	76.2 (73.7, 79.5)
IHD, mm	2.5 (2.0, 3.0)
tCD, mm	47.6 (45.7, 49.4)
cBPD, Z-Score	−2.3 (−2.8, −1.8)
bBPD, Z-Score	−2.2 (−2.9, −1.7)
IHD, Z-Score	0,3 (−0.3, 0.8)
tCD, Z-Score	−0.7 (−2.1, 0.0)
Impaired brain growth (cBPD), % (*n*) ^1^	67.8 (80/118)
Impaired brain growth (IHD), % (*n*) ^1^	8.5 (10/118)
Impaired brain growth (cBPD+IHD), % (*n*) ^1^	3.4 (4/118)

Data are % (*n*) or median (IQR) as appropriate. ^1^ Z-Score deviated >2 standard deviations from normative values. Abbreviations: FFM—fat-free mass, FM—fat mass, cBPD—cerebral biparietal diameter, bBPD—bone biparietal diameter, IHD—interhemispheric distance, tCD—transverse cerebellar diameter.

## Data Availability

The data presented in this study are available on request from the corresponding author. The data are not publicly available due to ongoing research.

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
