# Peer review of "Association between Fat-Free Mass and Brain Size in Extremely Preterm Infants"

_nutrients, 2021, doi:10.3390/nu13124205_

Round 1

Reviewer 1 Report

This is an interesting study aimed to assess the association between fat-free mass (FFM) and brain size in extremely preterm infants. A single point of assessment at term-equivalent age included measurements of body composition using air displacement plethysmography and of brain size measurement using magnetic resonance imaging.

Some issues need to be clarified or addressed.

Major queries:

  • FFM z-score was found to be independently and significantly correlated with higher cerebral biparietal diameter (cBPD) and transverse cerebellar diameter (tCD) z-scores, while fat mass (FM) z-score was independently and significantly correlated with lower cBPD and bone biparietal diameter (bBPD) z-scores. Associations of the brain size with the FM are as important as those found with the FFM, as both are dependent (eventually interrelated) on the nutritional support in preterm infants. Therefore, similar emphasis should be given to the associations found between both body compartments and brain size in the Abstract, Results, Discussion and Conclusion.
  • The Authors should explain or try to explain the mechanisms underlying the associations found of FFM and FM with specific brain size parameters.
  • The method used for recruitment of participants should be specified: consecutive cases? Random? By convenience?

Minor queries:

  • Lines 12, 41, and 181: I suggest replacing “for” with “associated” or “to support”, since typically the brain is not part neither of FFM nor of FM in bi-compartmental models commonly used to assess body composition in neonates.
  • Lines 44: “linear growth parameters” should be replaced with “anthropometric parameters”, since linear growth should be reserved to address increase in stature (length in this case)
  • Lines 138-142: statements should be uniform when describing results of associations of both body compartments with brain size. To be uniform with the statement regarding FFM associations (lines 138- …” (lines 140140), I suggest stating “FM z-score was significantly correlated with lower -141). Also in line 150 I suggest stating “FM z-score was independently and significantly… (lines 149-150).
  • Lines 152 and 54: As body compartments and brain size measurements were undertaken at a single point of assessment, a causal relationship cannot be established. Therefore, “impact” should be replaced with “association”.

Reviewer 2 Report

Study aim was to evaluate the association between body composition, linear growth parameters and brain size at term adjusted age in very low birth weight (VLBW) infants.

Study background / introduction need improvement. 1) Provide some elaboration on what constitute 'appropriate', 'inadequate' nutrition. 2) Authors gave the simplistic impression that growth failure in VLBW infants result from just inadequate nutritional intake whereas this is often more complicated and sometimes even enhanced nutritional strategies still result in abnormal body composition and disproportionate growth.  Site more of the previous studies relating 'optimal body composition' to 'normal brain growth and neurodevelopment' in VLBW infants. What did you do differently in this study? 

This is a single center retrospective study at a level IV NICU. Included infants < 28 weeks GA at birth. Authors need to comment on the infants with > grade 2 intraventricular hemorrhage (IVH); e.g. did any have post hemorrhagic hydrocephalus?

Authors found positive association between FFM Z-score and brain size Z-scores in this VLBW population, similar to previous findings and suggestion that FFM Z-scores rather than linear growth are more closely associated with brain growth. However, as indicated by the authors in the discussion, studies of the relationship between FFM, brain size / growth and neurodevelopmental outcomes have yielded conflicting results. Authors need to speculate on possible explanations for the inconsistent results in the discussion.

This study found length of parenteral nutrition to be significantly associated with FFM and brain size, contrary to findings in other studies. Authors should elaborate on this finding and offer possible explanation, if any, in the discussion. 

While the findings in this study are interesting, they are not new, and the real impact of the findings will become apparent when data on the neurodevelopmental assessment of the cohort become available.

Round 2

Reviewer 1 Report

The revised manuscript is substantially improved and should be accepted, provided minor queries are addressed.

There are still minor queries that need to be addressed:

  • Line 212: I suggest replacing “failure” with “restriction”
  • Lines 225: I suggest replacing “is” with “may be”. Currently, there is not sufficient evidence to state assertively that FFM is an indicator of brain growth
  • Line 231: “after term” should be replaced with “term-equivalent age”
